# Modeling Bounded Rationality in Multi-Agent Simulations Using Rationally Inattentive Reinforcement Learning

**Tong Mu**[*]                                                                    *tongm@cs.stanford.edu*
*Salesforce Research, Palo Alto, CA, USA*

**Stephan Zheng**                                                    *stephan.zheng@salesforce.com*
*Salesforce Research, Palo Alto, CA, USA*

**Alexander Trott**[*]
*Salesforce Research, Palo Alto, CA, USA*

**Reviewed on OpenReview:** *https://openreview.net/forum?id=DY1pMrmDkm*

## Abstract

Multi-agent reinforcement learning (MARL) is a powerful framework for studying emergent behavior in complex agent-based simulations. However, RL agents are often assumed to be rational and behave optimally, which does not fully reflect human behavior. In this work, we propose a new, more human-like RL agent, which incorporates an established model of human-irrationality, the Rational Inattention (RI) model. RI models the cost of cognitive information processing using mutual information. Our RIRL framework generalizes and is more flexible than prior work by allowing for multi-timestep dynamics and information channels with heterogeneous processing costs. We demonstrate the flexibility of RIRL in versions of a classic economic setting (Principal-Agent setting) with varying complexity. In simple settings, we show using RIRL can lead to optimal agent behavior policy with approximately the same functional form of what is expected from the analysis of prior work, which utilizes theoretical methods. We additionally demonstrate that using RIRL to analyze complex, theoretically intractable settings, yields a rich spectrum of new equilibrium behaviors that differ from those found under rationality assumptions. For example, increasing the cognitive cost experienced by a manager agent results in the other agents increasing the magnitude of their action to compensate. These results suggest RIRL is a powerful tool towards building AI agents that can mimic real human behavior.

## 1 Introduction

Multi-agent reinforcement learning (MARL) has shown great utility in complex agent-based model (ABM) simulations (e.g., in economics, games, and other fields (Zheng et al., 2020; Baker et al., 2020; Leibo et al., 2017; Yang et al., 2018). ABM is an influential field of research which examines various real-world systems through simulation to quantify the potential impacts of various policies (eg. market (Bozanta & Nasır, 2014) or environmental impact (Raihanian Mashhadi & Behdad, 2018) simulations). It has the promising potential to allow for better, more informed policies in critical applications areas, such as federal government economic policy (Farmer & Foley, 2009). Classically, in ABM, designers programmatically specify the behavioral rules of agents. This may be difficult in complex simulations, so in the MARL approach, designers instead specify the agents' objective functions and the reinforcement learning (RL) agents autonomously learn behaviors to optimize their objectives.

However, using rational RL agents is problematic when simulating systems with *human* agents. Rational agents execute the objective-maximizing behavior, which can differ greatly from established models of hu-

---

man decision-making. But humans often act non-rationally: they might not optimize perfectly for their objectives, and it is unclear to what degree human behavior can be described using a single (composite) scalar optimization objective. Such non-rational behavior is referred to as *bounded rationality.*

Prior economics literature has shown that modeling bounded rationality during decision-making gives results and implications that are significantly different from those obtained using rationality assumptions (Sims, 2003; Maćkowiak & Wiederholt, 2011; Jiang et al., 2019). Therefore, it is important to account for bounded rationality when simulating human(-like) agents. To address this issue, there has been a body of research in the economic literature that analyzes systems when human decision making follows established models of human-irrationality (Barazza & Strachan, 2020; Wu & Chen, 2014; Mullainathan, 2002; Huang et al., 2013; Ren & Huang, 2018), such as under the Rational Inattention model (Sims, 2003; Hoiles et al., 2020; Mackowiak et al., 2021; Jiang et al., 2019).

In the direction of this prior work, we study incorporating an explicit model of bounded rationality into MARL simulations. A key issue in prior work is that explicitly finding agent behaviors, given an operational definition of bounded rationality, can be a challenging learning and optimization problem. Here, reinforcement learning provides a powerful conceptual and algorithmic framework to model bounded rational behaviors. To this end, we introduce Rational Inattention Reinforcement Learning (RIRL), a MARL framework with agents that follow the well-established Rational Inattention (RI) model of bounded rationality (Sims, 2003; Mackowiak et al., 2021). The RI model attributes human irrationality to the *costliness of mental effort*, e.g., attention, required to identify the optimal action. Mathematically, the RI framework measures these costs as the *mutual information* (MI) between the variables considered by the decision process and the decisions ultimately made. This captures the intuition that a policy has a higher cognitive cost if its execution requires more information about the state of the world and thus more attention.

MI-based rewards have been used in RL, (e.g. Leibfried & Braun (2016); Leibfried et al. (2018); Grau-Moya et al. (2018); Leibfried & Grau-Moya (2020) use RI-style MI costs to regularize learning) but not to model sub-optimal behavior. When used to model sub-optimal behavior, RIRL can rationalize seemingly sub-optimal behavior by including this cognitive cost in the reward function, i.e., by adding the MI cost(s). That is, the "rational" behaviors of an RIRL-actor can mimic human-like bounded rationality.

## 1.1 Contributions

In this paper, we introduce a novel and flexible policy class for analyzing multi-agent systems with agents that are boundedly rational. The main contributions of this work are:

1. We propose RIRL, a framework for modeling bounded-rationality which can analyze complex scenarios, not previously possible. Our method allows for the analysis of settings where different information has different observation or processing costs and multi-timestep settings.

2. We validate RIRL's modeling capabilities in classical economic settings, where the majority of prior work focused on theoretically solving for the agent's behavior. We demonstrate in a simpler, previously studied setting that our results match the expected functional form. Additionally in a more complex, theoretically intractable setting, our method is able to find equilibria of agent behavior that are different from rational equilibria and can be used to provide detailed analysis of these phenomena.

**Modeling bounded rationality in complex environments.** RIRL extends the single-timestep framework proposed by Peng et al. (2017), which decomposes decision-making into stochastic perception followed by stochastic action, each subject to its own MI cost. RIRL provides a novel boundedly-rational policy class that greatly expands this framework to multiple information channels with heterogeneous costs, hidden-state policy models, and sequential environments. To achieve this increased modeling power, our architecture uses methods significantly different from prior work, such as using deep mutual information estimation modules to allow for multiple channels and using an LSTM to account for multiple timesteps. This allows RIRL to analyze settings with rich cognitive cost structures, e.g., when information about state variables have

different observation difficulty. For example, in hiring, a candidate's past job performance may be more relevant but harder to evaluate than her employment history.

**Evaluation in the Principal-Agent setting.** To illustrate our framework's modeling abilities, we evaluate RIRL in classical economic settings, specifically two *Principal-Agent* (PA) problems (Grossman & Hart, 1992; Sannikov, 2008; Haubrich, 1994) and we consider when the Principal is boundedly rational. Typical economics research rely on theoretical analysis to analyze these settings. We use RIRL to model information asymmetry between the Principal and the Agents they are incentivizing, i.e., it is hard for the Principal to obtain information about the Agents. Real-world PA experiments have shown that bounded rationality is key to explaining marked deviations between equilibria reached by human participants and theoretical predictions (Erlei & Schenk-Mathes, 2017). Prior work considered bounded rationality assumptions but found it to be difficult to analyze theoretically (Mirrlees, 1976); research in this direction is sparse.

We show that RIRL allows us to analyze generalized PA problems that are analytically intractable, such as a sequential PA problem with multiple Agents and heterogeneous information channels. Across all settings, we observe that equilibrium implications depends strongly on the cost(s) of attention and differs from that under rational assumptions sometimes in ways that are hard to intuitively predict. We observe that increasing Principal inattention can either increase Agent welfare due to increased compensation or decrease Agent welfare due to encouraging additional work. Compared with rational agents, we additionally observe the emergence of different strategies, such as Agents choosing to misrepresent their ability, classically referred to as *signaling* (Spence, 1973). Our results show that RIRL can be a powerful tool to model boundedly rational behavior and analyze its emergent consequences in multi-agent systems.

## 2 Related Works

**Multi-Agent Reinforcement Learning for Agent-Based Simulation.** Agent-Based Models (ABM) are a popular tool used to study real-world systems (e.g., organizations or market systems) and discover emergent behavior through simulation with fixed or manually-specified agent behaviors (Bonabeau, 2002). Instead, Multi-Agent Reinforcement Learning (MARL)-based simulations use RL agents which autonomously learn utility-maximizing behavior, so designers do not need to specify behavioral rules. MARL has been used to study tax policy (Zheng et al., 2020; 2021; Trott et al., 2021), games (Baker et al., 2020), and social dilemmas (Leibo et al., 2017) among others (Yang et al., 2018). However, RL agents are mostly assumed rational which contradicts how humans make decisions. Some recent work considers bounded rationality in MARL by accounting for cognitive limitations when reasoning about the behaviors of other agents (Evans & Prokopenko, 2021; Wen et al., 2019; Łatek et al., 2009). We model a complementary source of bounded rationality: the cognitive costs of attending to all available information when forming a decision.

**Models of Irrationality.** Behavioral economics has shown extensively that human decision-making is not fully rational but instead features many cognitive biases (Caverni et al., 1990). Bounded rationality (Simon, 1957) attributes these human irrationalities to resource limitations, e.g., bounded cognitive capabilities or costliness of using (more) time to make decisions. Simon (2000) gives a high-level overview of different ways to implement bounded rationality, but did not yet construct explicit solutions for (optimal) agent policies (under bounded rationality). Our work relates to this by learning explicit solutions in a setting with multiple strategic agents. As another example, Hernandez & Ortega (2019) studied bounded rationality in organizations and its impact on growth.

Rational Inattention (RI) is a well-established model of bounded rationality which models the cognitive cost of decisions as the mutual information between variables relevant to the decision and the decision itself (Sims, 2003; Mackowiak et al., 2021). RI has been tested in real world experiments (Dean & Neligh, 2017) and used to model human behavior in a wide variety of domains (Hoiles et al., 2020; Mackowiak et al., 2021). Closest to our work, Jiang et al. (2019) study a multi-agent system for traffic route choice under RI but where the agents do not react to each other's actions. Another line of work uses human behavioral data with deep learning to model human cognition (Kubilius et al., 2019; Battleday et al., 2017; Ma & Peters, 2020) or predict human behavior (Bourgin et al., 2019; Kolumbus & Noti, 2019; Hartford, 2016). In contrast, our

ABM approach examines the implications of RI in complex settings where experimental data are presently unavailable.

**Mutual information and rational inattention.** MI has been extensively studied in the intrinsically motivated RL literature for curiosity-driven exploration and unsupervised skill and option discovery (Still & Precup, 2012; Campos et al., 2020; Mohamed & Rezende, 2015; Gregor et al., 2016; Eysenbach et al., 2019), often with differing techniques used to measure MI. Recently MI-based rewards have been used to regularize exploration (Grau-Moya et al., 2018; Leibfried & Grau-Moya, 2020). They have also been used to regularize policy learning (Malloy et al., 2021) to learn better policies. However this work differs as it does not consider modeling bounded rationality for more realistic MARL simulations. Comparatively, very little work has considered MI-based rewards for modeling boundedly rational behavior (Ortega et al., 2015; Peng et al., 2017). Genewein et al. (2015) introduced a mutual information between action and world state. However, it focuses on the single-agent setting. In our paper, we consider multiple (strategic agents), which significantly complicates finding the optimal policy for the bounded rational agent, and the corresponding best-response of the other agent(s). To our knowledge, there is with no prior work considering the domain of multi-agent simulations.

For example, our policy class considers allowing a rich set of behavior to be modeled, such as different attention costs on different parts of the observation space. Additionally, due to the difference in settings, we also draw different conclusions, focusing on the implications of modeling human irrationality such as which agents benefit and how metrics, like equity, evolve.

**Principal-Agent problems.** In addition to our guiding example, Principal-Agent problems have been used to describe various settings, including politics and insurance (Miller, 2005; Grossman & Hart, 1992). Prior economics literature mainly use analytical methods and narrow modeling assumptions, e.g., that all stochasticity follows Brownian motion (Sannikov, 2008) or certain separability conditions are satisfied (Grossman & Hart, 1992). Generally, Principal-Agent settings fall under the category of Stackelberg games where it has been shown computing optimal strategies can be NP-hard (Korzhyk et al., 2010). Instead, MARL can study complex setups that are analytically intractable. Shu & Tian (2019); Shi et al. (2019); Ahilan & Dayan (2019) studied coordinating cooperation in a Principal-Agent model, with an RL Principal learning to incentivize Agents to achieve an overall goal. However, they do not consider bounded rationality.

**Bounded rationality and reinforcement learning.** Kokolakis et al. (2021) models bounded rationality using level-$k$ policies instead of perfect rationality, which converge to the Nash equilibrium as $k$ increases. Erevl & Roth (2002) use RL to model bounded rationality agents in iterated prisoners dilemmas, extending prior analytical models in game theory. Duéñez-Guzmán et al. (2021) studies how statistical discrimination may arise in agents due to (limitations on) information processing using a theoretical model and RL agents.

## 3   Preliminaries

We consider multi-agent simulations in the form of partially-observable Markov Games (MGs), formally defined by $(S, A, r, \mathcal{T}, \gamma, O, \mathcal{I})$ (Sutton & Barto, 2018). Here $S$ is the state space of the game, $A$ is the combined action spaces of the actors, and $\mathcal{I}$ are actor indices. Generally, the full state is not observable to each actor (for example, actors may not be able to observe the full policy of other actors, and only the history of actions the other actors took). We use $o_i = O(s, i)$ to denote the portion of the game state $s$ that actor $i$ can observe and use for their policy. In addition, $o_i$ may include a (possibly learnable) encoding of the observation history. Each game episode has a horizon of $T \geq 1$ timestep(s). Each timestep $t$, actor $i$ selects action $a_{i,t}$, sampling from its policy $\pi_i(a_{i,t}|o_{i,t})$. Given the sampled actions, the transition function $\mathcal{T}$ determines how the state evolves. Each actor's objective is encoded in its reward function $r_i(s, \boldsymbol{a})$, where boldface denotes concatenation across actors.[1]

---

[1] **Reward vs utility:** In the economics literature an agent's reward is often modeled by its *utility* (its measure of happiness) and denoted by $u$. Similarly, rewards and utilities model *incentives* that motivate an agent's behavior. In this work, we will use "reward", denoted by $r$, throughout to refer to these concepts.

Traditionally, prior work considers the case where the actor optimizes its policy $\pi_i$ to maximize its $\gamma$-discounted sum of future rewards:

$$\pi_i^* = \arg\max_{\pi_i} \mathbb{E}_{\boldsymbol{\pi},\mathcal{T}} \left[ \sum_t \gamma^t r_i(s, \boldsymbol{a}) \right]. \tag{1}$$

Using the subscript notation where subscript $i$ (eg. $a_{t;i}$ denotes the vector element relating to actor $i$ and $-i$ (e.g. $\boldsymbol{a}_{t;-i}$) indicates a vector with elements that are *not* pertinent to actor $i$.

$$\pi_i^* = \arg\max_{\pi_i} \mathbb{E}_{\pi_i,\boldsymbol{\pi}_{-i}^*,\mathcal{T}} \left[ \sum_t \gamma^t r_i(s_t, a_{t;i}, \boldsymbol{a}_{t;-i}^*) \right], \tag{2}$$

$$\text{s.t. } a_{t;j}^* \sim \pi_j^*(.|o_t), \quad \pi_j^*(.|o_t) = \arg\max_{\pi_j} \mathbb{E}_{r_j,\boldsymbol{\pi}_{-i},\mathcal{T}} \left[ \sum_{v \geq t} \gamma^v r_j\left(s_v, a_{v;i}, \boldsymbol{a}_{v;-i}\right) \middle| a_{t;i} \right], \quad \forall j \neq i. \tag{3}$$

Equation 2 states that the actor optimizes its return assuming the other actors play a best response, i.e., use a policy $\pi_j^*$ that maximizes their return.

### 3.1 Modeling Bounded Rationality using Rational Inattention

While RL can be used to discover (approximately) reward-maximizing policies, such rational behavior fails to account for characteristic *irrational* human behavior. Rational Inattention models bounded rationality using a modified objective that includes a cost to the mutual information $I(a_i; o_i)$ between the (observable) state of the world $o_i$ and the actions $a_i \sim \pi_i(\cdot|o_i)$. This definition captures the intuition that if the agent puts in more effort to pay attention to $o_i$, its action $a_i$ likely becomes more correlated with the observation $o_i$, and thus Mutual Information is high.

We optimize for the policy of the rationally-inattentive objective:

$$\pi_i^\dagger = \arg\max_{\pi_i} \left( \mathbb{E}_{\boldsymbol{\pi}} \left[ \sum_t \gamma^t \left( r_i(s_t, \boldsymbol{a}_t) - \lambda I(a_{i,t}; o_{i,t}) \right) \right] \right). \tag{4}$$

Note that this is equivalent to learning the optimal policy for an adjusted reward function: $r_i^\dagger(s_t, \boldsymbol{a}_t) = r_i(s_t, \boldsymbol{a}_t) - \lambda \tilde{I}(a_{i,t}; o_{i,t})$, where

$$\tilde{I}(a_{i,t}; o_{i,t}) = \log \frac{p(a_{i,t}, o_{i,t})}{p(a_{i,t})p(o_{i,t})} \tag{5}$$

is a Monte Carlo estimate of $I(a_i; o_i)$, and $\lambda$ is the cost per bit of information. Here $p(a_i, o_i)$, $p(a_i)$, and $p(o_i)$ are the joint and marginal distributions over $a_i$ and $o_i$ induced by the environment and policies $\boldsymbol{\pi}$.

To build intuition for the use of $I$, imagine the extreme case where the actor pays *no* attention to $o_i$ (e.g. the actor selects actions uniformly at random). The probability of a given action being selected wouldn't depend on the observation and the MI cost $I(a_i; o_i)$ would be 0. On the other hand, if the action correlates with the observation, the actor is assumed to have paid attention to $o_i$ in order to select $a_i$ and $I > 0$. From a learning perspective, the MI cost acts as a regularizer which encourages simpler policies, i.e., where actions are less dependent on the precise state of the world and/or history of observations. Rationally inattentive actors therefore are incentivized to select which information they use, making inattention an *endogenous* source of information asymmetry. From an economic perspective, the attention cost may model limits on information processing and hence encourage different types of actor strategies. Alternatively, attention costs could reflect actual expenditures, i.e., monitoring costs.

### 3.2 Multiple Information Channels

To model more general forms of rational inattention, we extend our framework to support:

1. multiple channels of information with heterogeneous cognitive costs, and

2. modeling which observation channel an actor pays attention to.

For example, when buying a used car it is easier to see car prices (which might still take time) than ascertaining the actual condition of the car.

As such, we first extend the *action-perception decoupling* strategy of Peng et al. (2017) to model $\pi(a|o)$ as a stochastic perception module $q(y|o)$ (encoder) followed by an action module $\omega(a|y)$ (decoder). The RI reward then becomes:

$$r^\dagger = r(s, a) - \lambda_q I_q(y; o) - \lambda_\omega I_\omega(a; y), \tag{6}$$

where we omitted subscripts for clarity. In the above, recall $s$ denotes the true state of the world and $o$ denotes the portion observable to the agents. Parameters $\lambda_q$ and $\lambda_\omega$ control the amount of inattention simulated. This decomposes the attention cost into: (1) the cost of paying attention to the observation $o$, and (2) the cost of using that information to select an action.

Second, we extend the encoder to capture attention costs that depend on $M$ parallel observations $\{o^1, \ldots, o^M\}$, giving an RI reward function:

$$r^\dagger = r(s, a) - \sum_{i=1}^{M} \lambda^i I(y^i; o^i) - \lambda_\omega I_\omega(a; [y^1, \ldots, y^M]), \tag{7}$$

where brackets denote concatenation, and each channel has its own attention cost $\lambda^m$ of observing $o^m$.

## 4 Modeling and Training Boundedly Rational Actors with RIRL

We now describe a practical approach to estimating the attention cost in Equation 4 and solving the optimization problem in Equations 2 and 3.

These optimization problems present several technical challenges: (i) Finding the optimal policy is challenging in a sequential setting and with multiple actors. (ii) Computing the true mutual information cost $I$ is computationally expensive.

Our framework RIRL solves the policy optimization problem using reinforcement learning (RL), while we propose a sampling-based estimator for $I$ that works well in practice.

### 4.1 Mutual Information Estimation

The key challenge in computing $I$ is that computing $p(o)$ requires full knowledge of the dynamics of the Markov Game, while computing $p(a)$ requires marginalizing over all observations (which can be intractable when, e.g., the observation space is high-dimensional). Hence, we use a Monte Carlo estimate $\tilde{I}_\pi(a_t; o_t)$ of $I(a; o)$, using data $\{(a, o) : a \sim \pi\}$ collected while executing policy $\pi$. Note an unbiased estimator has $I = \mathbb{E}_\pi[\tilde{I}]$. The reward for a rationally inattentive actor is then computed as $r_{t;i}^\dagger = r_{t;i} - \lambda \tilde{I}_{\pi_i}(a_{t;i}; o_{t;i})$.

**Estimating Mutual Information using Discriminators.** Given a pair $(a, o)$ for a single agent (omitting subscripts for clarity), we estimate $\tilde{I}_\pi(a; o)$ from the ratio between $\log p(a, o)$ (the log-odds under the joint distribution) and $\log p(a)p(o)$ (the log-odds under the factorized distributions). This ratio can be estimated using a *discriminator model* $D_\pi(a, o)$ that learns to classify whether the sample $(a, o)$ came from the joint or factorized distribution. We generate samples from $p(a, o)$ by sampling trajectories using $\pi$, while samples from $p(a)p(o)$ can be generated by creating random pairs $(a, o)$ from a batch of sampled trajectories. This approach is inspired by the structure of Generative Adversarial Networks Goodfellow et al. (2014), which use discriminators to learn to synthesize images that are almost indistinguishable from real images. This technique for MI estimation can be applied for any pair of variables within some generative process. We leverage this generality throughout RIRL.

Our framework is compatible with any MI estimator. Our approach is simple and effective, although other MI estimation techniques have been proposed, e.g., Belghazi et al. (2018).

**Limited variance of our mutual information estimator.** In principle, this sample-based estimator could have high variance, but in practice, even using a single sample to compute $\tilde{I}$ is effective. In particular, estimating $I$ using data sampled from the RL policy works well.

There are multiple reasons for this. First, we train a discriminator to estimate the ratio $\frac{p(a,o)}{p(a)p(o)}$, not the individual probabilities itself. Second, the RL policy typically changes slowly during training. Hence, there is only a small amount of distribution shift in the rollout samples across training steps. So using a single example could be high variance, but training still works when training the discriminator slowly enough. Third, the pair $(a, o)$ of a randomly sampled $a$ and a randomly sampled $o$ can be taken from a batch of trajectories that were collected under different states $s$, because such randomly sampled pairs are more likely to be independent across trajectories as the policy changes.

As such, if we sampled many trajectories from the same policy $\pi$, and given a discriminator with enough model capacity, it should have the capacity to learn the true odds-ratios. With a smaller number of samples, training the discriminator needs to be done slowly enough, but in practice, our approach is sufficient to learn meaningful MI estimates, i.e., that sufficiently distinguish independent vs dependent samples.

## 4.2 Recurrent Policies for Sequential Settings and Heterogeneous Attention Costs

In addition, we aim to apply our framework in the sequential setting and with heterogeneous attention costs. To do so, we use a recurrent model that can summarize historical observations. In particular, such models can learn to strategically allocate attention over time.

Before describing details, we provide the following rough sketch of our policy architecture: (i) The observation is first separated into distinct "channels". (ii) For each channel, there is an encoding module which converts its portion of the observation into a noisy encoding. (iii) The encodings from each channel are combined and then fed into a recurrent neural network model to update its recurrent state. (iv) The combined encodings and the updated recurrent state are fed into a stochastic action module to produce a probability distribution over actions and sample accordingly.

### 4.2.1 Encoder Modules

For each channel, we use the reparameterization trick Kingma & Welling (2013) to compute the mean and standard-deviation of a Gaussian distribution over $y_t^m$, which is the noisy encoding of $o_t^m$:

$$\mu_t^m, \sigma_t^m = q^m(o_t^m, h_t), \tag{8}$$
$$y_t^m = o_t^m + \mu_t^m + \sigma_t^m \cdot \epsilon_t^m, \quad \epsilon_t^m \sim \mathcal{N}(\mathbf{0}, \mathbf{1}). \tag{9}$$

Here $h_t$ is the recurrent state of the recurrent model (see below), and $\epsilon_t^m$ is a random sample from a spherical Gaussian with dimensionality equal to that of $y^m$ and $o^m$.

This decomposition allows us to separate out attention costs for distinct channels of information. In other words, for each channel $m$ we want to add an attention cost proportional to the amount of information about $o^m$ encoded in $y^m$. To that end, analogous to Section 4.1, for each channel $m$ we also train a discriminator $D_q^m(y_t^m, [o_t^m, h_t])$ used to estimate $\tilde{I}_m(y_t^m; [o_t^m, h_t])$, where brackets denote concatenation.

### 4.2.2 Recurrent Stochastic Action Module

The full encoding $y_t = \left[y_t^1, \ldots, y_t^M\right]$ of $o_t$ concatenates all $M$ encoder samples. We introduce recurrence by layering a recurrent neural network model that takes $y_t$ and $h_t$ as inputs and outputs an updated recurrent state $h_{t+1}$. In our experiments, we use LSTMs (Hochreiter & Schmidhuber, 1997), although many other variations exist in the machine learning literature. The encoding $y_t$ and recurrent state $h_{t+1}$ are used as inputs to the decoder $\omega(a_t|y_t, h_{t+1})$ which outputs a distribution over actions, from which $a_t$ is sampled. As with the encoders, we can train a discriminator $D_\omega(a_t, [y_t, h_{t+1}])$ used to estimate $\tilde{I}_\omega(a_t, [y_t, h_{t+1}])$, introducing an additional attention cost capturing how much of the encoded information is used to select an action.

### 4.3 Optimizing with Reinforcement Learning

Given the model structure discussed previously, we now solve the optimization problem (Equation 2) using reinforcement learning (RL) Sutton & Barto (2018). Denoting the weights of a policy by $\theta$, we update $\pi(a|o;\theta)$ using policy gradients (Williams, 1992):

$$\Delta\theta \propto \mathbb{E}_{\boldsymbol{\pi},\mathcal{T}}\left[\sum_t \nabla_\theta \log \pi(a_t, y_t|o_t, h_t, h_{t+1}; \theta) R_t^\dagger\right], \quad R_t^\dagger = \sum_{k=0}^{T-t} \gamma^k r_{t+k}^\dagger, \tag{10}$$

$$\log \pi(a_t, y_t|o_t, h_t, h_{t+1}) = \log \omega(a_t|y_t, h_{t+1}) + \sum_{m=1}^M \log q^m(y_t^m|o_t^m, h_t), \tag{11}$$

$$r_t^\dagger = r(s_t, \boldsymbol{a}_t) - \lambda_\omega \tilde{I}_\omega(a_t; [y_t, h_{t+1}]) - \sum_{m=1}^M \lambda_m \tilde{I}_m(y_t^m; [o_t^m, h_t]). \tag{12}$$

For further details on training and model implementation, see the Appendix.

## 5 Validating RIRL in Principal-Agent Problems

We demonstrate RIRL's modeling capabilities in two Principal-Agent (PA) problems of varying complexity. We specifically consider manager-workers relations, and in both settings, a boundedly rational manager (the Principal) decides a compensation structure for the workers (the Agents). In general, labor is costly to the Agents but beneficial to the Principal, while the Agents benefit from income that is costly for the Principal to provide. Therefore, the Principal aims to find a wage schedule $\mathcal{W}$ that maximizes its profits. PA problems often consider how *information asymmetry* between the Principal and Agent(s) influences equilibrium schedules. RIRL can analyze a meaningful extension in which we model how the *cost of information* influences equilibrium schedules, i.e., where the Principal can spend effort to reduce information asymmetry. We focus on studying bounded rationality in the principal as is common in prior work (Mirrlees, 1976).

We first study a single-timestep PA setting with a Principal that optimizes a wage schedule for each level of Agent output, subject to a cognitive cost for observing output. Because some aspects of the solution are tractable with prior analytical methods, this simple setting allows us to draw comparisons and provides a useful validation of RIRL in a multi-Agent game. However note, even in this simple setting, aspects of the solution (such as the constants) are not tractable analytically. Secondly, we apply RIRL to analyze a complex sequential PA problem outside the scope of analytical methods of prior work that involves multiple Agents and heterogeneous information channels.

Demonstrating the importance of modeling bounded rationality, our experimental results were able to uncover several non-trivial insights when bounded rationality is assumed that are hard to intuitively predict. Specifically, in our PA settings, insights include (1) Agent reward can increase with principal inattention and a more rational Agent benefits more from Principal inattention due to the Principal setting higher pay schedules. To our knowledge, no prior work has identified the consequences of bounded rationality in both the Principal and Agent. (2) In the case of multiple Agents, a boundedly rational Principal can induce Prisoner's Dilemma like incentives for the Agents, reducing the welfare of all agents. We discuss these insights along with others in detail below.

### 5.1 Principal-Agent with a Single Agent, Single Timestep

We first evaluate in a PA setting with a two actors (the Principal and a single Agent), following classic work (Mirrlees, 1976; Holmstrom & Milgrom, 1987; Spremann, 1987; Haubrich, 1994). When modeling economic behavior, the reward is the (marginal) reward $r$, an abstract measure of happiness. We use such terminology throughout the sections. The Agent's labor output $z \sim h(e)$ is a stochastic function of its effort action $e$. The Agent chooses how much to work based on its wage schedule $\mathcal{W}$, which yields wage $w$ as a stochastic function of output $z$, i.e., $w \sim \mathcal{W}(z) = \mathcal{N}(\mu_z, \sigma_z)$. Here, the stochasticity in the pay schedule aims to model the Principal's uncertainty over $z$, e.g., due to limited attention. By assumption, the Agent accurately

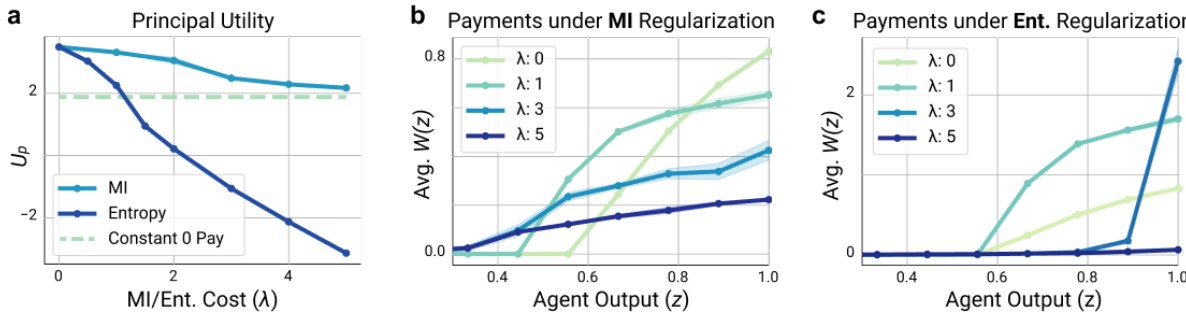

Figure 1: Bandit Experiment Results. All results are averaged over 5 random seeds and plotted with 95% confidence intervals. **(a)** Comparing MI and Entropy regularization on Principal reward. The constant 0-pay schedule provides a meaningful lower bound. **(b, c)** Pay schedule means under MI (b) and Entropy (c) regularization (pay schedule standard deviations are plotted in the Appendix).

anticipates the shape of the wage schedule $\mathcal{W}$ when making its decisions, e.g., based on prior experience or reputation. As such, the RIRL-Principal sets the wage schedule:

$$\mathcal{W}^* = \arg\max_{\mathcal{W}} \left[\mathbb{E}\left[r_p(w,z)\right] - \lambda I_{\mathcal{W}}(w;z)\right]_{w\sim\mathcal{W}(z), z\sim h(e), e\sim\pi_a^\beta(e|w)}. \tag{13}$$

Here, $\lambda I_{\mathcal{W}}(w;z)$ is the attention cost for $\mathcal{W}$.

For the Agent, we use a soft-Q policy $\pi_a^\beta(e|w) = e^{\beta r_a(e|w)}/Z$ with normalization $Z$, where $r_a(e|w)$ denotes the *expected* reward of effort $e$ given wage $w$, sampled from the schedule $\mathcal{W}$. Note that $\pi_a^\beta$ can be calculated directly. Importantly, *the soft-Q policy models a form of bounded rationality*, i.e., the log-odds of effort $e$ is proportional to its expected reward, with higher $\beta$ making "better" actions $e$ more likely. Hence, low $\beta$ corresponds to less-than-rational behavior.

The reward of the Principal ($r_p$) and Agent ($r_a$) follow standard economic reward functions:

$$r_a(w,e) = \underbrace{\texttt{CRRA}(w;\rho)}_{\text{Income reward}} - \underbrace{e}_{\text{Work Disreward}} \quad \text{and} \quad r_p(w,z) = \underbrace{z}_{\text{Profit}} - \underbrace{w}_{\text{Amount Paid}} \tag{14}$$

Here $\texttt{CRRA}$ is the concave Constant Relative Risk Adverse function (Pratt, 1978); the risk aversion parameter $\rho$ sets its concavity (we use $\rho = 2$). This models diminishing returns with income increases. More environment details are in the Appendix. We use $N$-sample policy gradients on estimates of Equation 13 to learn the parameters $\mu_z$ and $\sigma_z$ of $\mathcal{W}^*$

$$r_{\mathcal{W}}^\dagger = \frac{1}{N}\sum_{i=1}^N \left[r_p(w_i,z_i) - \lambda\tilde{I}_{\mathcal{W}}(w_i;z_i)\right], \quad w_i\sim\mathcal{W}(z_i), z_i\sim h(e_i), e_i\sim\pi_a^\beta(e|\mathcal{W}_i). \tag{15}$$

**Results.** Figure 1 shows the effects of the Principal's inattention on simulated outcomes. We also compare RIRL outcomes against those found including entropy-based rewards. MI-based and entropy-based regularization are closely related (Grau-Moya et al., 2018; Leibfried & Grau-Moya, 2020), with subtle but important differences[2]. Nevertheless, they yield markedly different outcomes.

To highlight an important difference, consider the constant 0-pay schedule. This provides no incentive for the Agent to work and costs the Principal nothing, and it therefore provides a reasonable lower bound on $r_p$. This lower bound should hold under bounded rationality, since no attention is required when $\mathcal{W}$ treats all outputs identically. Indeed, under RIRL, $r_p$ approaches this lower bound as increasing $\lambda$ leads the Principal to trade more profitable $\mathcal{W}$'s for ones with smaller demands on attention (Figs. 1a, b, 2c). In contrast, under entropy regularization, increasing $\lambda$ quickly yields $\mathcal{W}$'s that violate the $r_p$ lower bound (Fig. 1a, c).

---

[2]Under some conditions, they are identical. However, the key difference is that entropy regularization penalizes the policy for deviating from a *fixed, uniform* prior, whereas RI-style MI regularization penalizes deviations from an *optimal* prior.

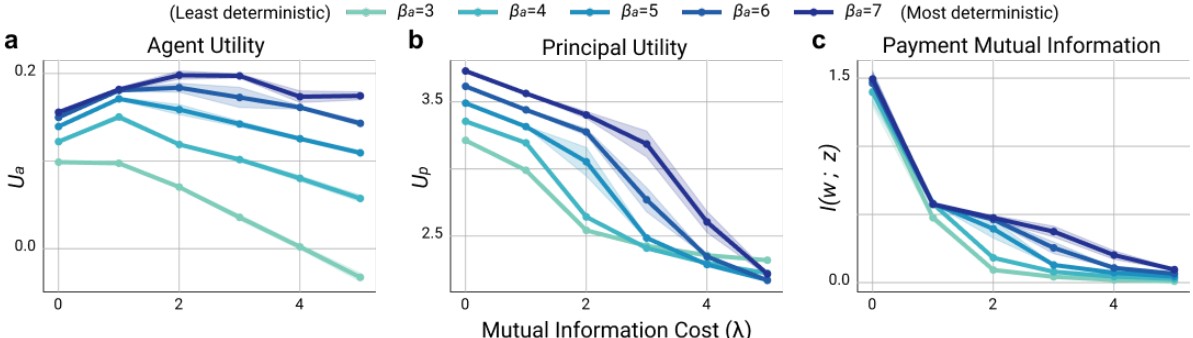

Figure 2: Additional Bandit Experiment Results. All results are averaged over 5 random seeds and plotted with 95% confidence intervals. Results are shown for each level of MI cost ($\lambda$, x-axis) across multiple levels of Agent policy temperatures ($\beta$, color). **(a, b)** Agent and principal reward. **(c)** Mutual Information between output $z$ and wage $w$.

Figure 2 shows how Agent and Principal reward and $I(w; z)$ change as a function of $\lambda$, across a range of Agent $\beta$'s. Note that higher $\beta$ increases the odds that the Agent selects the optimal action. This reveals an interesting multi-Agent interaction. As before, we observe that the Principal's reward $r_p$ (without the attention cost) decreases with increasing $\lambda$. Furthermore, $r_p$ is generally lower when $\beta$ is lower (Fig. 2b), as the Principal must pay more to influence $\pi_a^\beta$ towards different actions.

Interestingly, we observe that *the Agent's reward $r_a$ can actually increase when the Principal is inattentive*, and that higher $\beta$ tends to increase the beneficial range of inattention (Fig. 2a). Lastly, $\beta$ determines how the Principal responds to increasing attention costs. Agents that behave optimally more often (higher $\beta$) incentivize the Principal more to pay attention (Fig. 2c).

**Comparison With Prior Literature.**   For this setting, Mirrlees (1976) found the theoretically optimal pay schedule should be of the form $\mathcal{W}(z) = \max(Az + B, C)^{\frac{1}{\rho}}$. While they assume a slightly different model of costly attention[3], the emergent pay schedules match closely with this work: when fitting $A$, $B$, and $\rho$ to our data, we measure $r^2 = 0.99$ for all $\beta$. The best fit $\rho$ was close to the theoretical value for some values of $\beta$ (true $\rho$: 2, best fit $\rho$: 2.19 for $\beta = 5$). Interestingly, we do observe that the best fitting $\rho$ tends to increase with $\beta$ in our model. Notably, the influence of Agent sub-optimality was outside the scope of this prior work. We are able to observe this relationship between $\rho$ and $\beta$ through the modelling complexity enabled by our RIRL framework.

### 5.2   Principal-Agent in the Sequential Multi-Agent Setting

We show that RIRL enables modeling sequential Principal-Agent problems with multiple Agents with $T > 1$ timesteps. We consider horizons $[2, 10]$ and teams of $n_a = 4$ Agents. We assume there are $K = 5$ possible Agent abilities $k$. The Principal cannot see the Agents' abilities. Each Agent's ability is sampled randomly at the start of each episode. At each timestep, Agent $i$'s output is:

$$z_{i,t} = h_{i,t} \cdot (\nu_i + e_{i,t}), \tag{16}$$

where it works $h_{i,t}$ hours and exerts effort $e_{i,t}$. The Principal moves first and sets a wage $w_{i,t}$. Each Agent moves second: it knows $w_{i,t}$ before choosing $h_{i,t}$ and $e_{i,t}$, and earning income $w_{i,t} \cdot h_{i,t}$. As before, Principal reward $r_p$ measures profit. We define $r_a$ following standard reward functions, where the optimal $h$ increases with $w$. As a consequence of this configuration (demonstrated below), the profit-maximizing wage $w_i$ for

---

[3]Their Principal pays a cost to reduce the *noise* added to its observed output, as opposed to our MI cost; see the Appendix for full details.

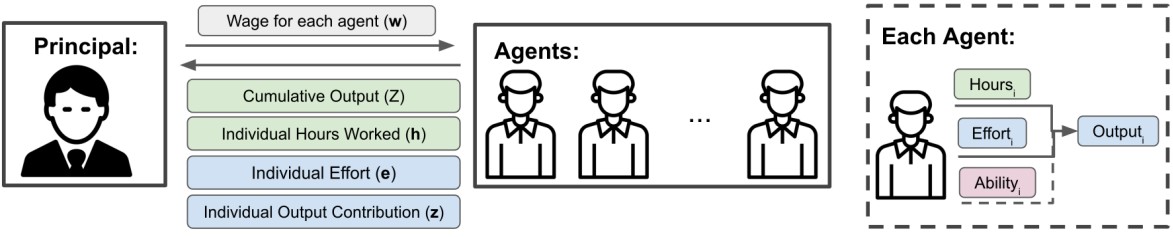

Figure 3: A depiction of a single timestep in an episode of our Sequential Multi-Agent Setting. Green variables (such as cumulative output) are not costly while blue variables (such as individual output) are costly for the principal to observe. The red variable (Ability) cannot be observed.

Agent $i$ increases with its ability $\nu_i$. The Agent reward is:

$$U_a(w, h, e) = \underbrace{\texttt{CRRA}(w \cdot h; \rho)}_{\text{Income reward}} - \underbrace{c_l h^\alpha (1 + e)}_{\text{Work Disreward}} , \quad \underbrace{U_p(\boldsymbol{w}, \boldsymbol{h}, \boldsymbol{z})}_{\text{Profit}} = \underbrace{\sum_{i \in [n_a]} z_i}_{\text{Revenue}} - \underbrace{\sum_{i \in [n_a]} w_i h_i}_{\text{Wages Paid}}, \quad (17)$$

where $\rho, c_l$ and $\alpha$ are constants governing the shape of $r_a$.

**Attention Costs.** Because a strategic Principal must *infer* private Agent features, e.g., ability, its equilibrium behavior depends on any inference costs it experiences, e.g., attention costs. Note that, unlike in the above bandit setting, here we train *both* the Principal and the Agent policies, each of which is modeled using the RIRL-actor architecture (Section 4). However, to isolate and explore the effects of distinct Principal attention costs, we do not impose any attention costs on the Agents. Therefore, the Agents' reward is only their reward $r_{i,t} = U_a(w_{i,t}, h_{i,t}, e_{i,t})$.

For the Principal, we model $M = 3$ information channels: one is "easy" and low-cost to observe and two are "hard" and high-cost. The low-cost channel $o_p^f$ includes information that we regard as freely available ($\lambda^f = 0$), e.g., the time $t$, the hours worked $\boldsymbol{h}$ (workers often fill out timesheets which makes $h_i$ easy to see), and the *total* output, $Z = \sum_{i \in [n_a]} z_i$ (managers can see the final result). However, it is high-cost to see *individual* contributions: we use a high-cost channel $o_p^e$ for efforts $\boldsymbol{e}$, and a high-cost channel $o_p^z$ for outputs $\boldsymbol{z}$. This models a Principal who can spend time and attention to observe individual Agents to reduce uncertainty about their true ability, e.g., their working styles and productivity (Figure 3). Modeling the attention cost of output and effort separately lets us study the effects and interactions of unequal observation costs. The Principal's reward is

$$r_{p,t}^\dagger = r_p(\boldsymbol{w}_t, \boldsymbol{h}_t, \boldsymbol{z}_t) - \underbrace{\lambda^z \tilde{I}(y_t^z; \boldsymbol{z}_t)}_{\text{Individual Output and}} - \underbrace{\lambda^e \tilde{I}(y_t^e; \boldsymbol{e}_t)}_{\text{Effort Perception Cost}} \quad (18)$$

Thus, the Principal's bounded rationality is modeled through the cost to get information about effort and individual outputs. The Principal's RIRL-actor architecture would also let us use attention costs $\tilde{I}(y_t^f; o_t^f)$ and $\tilde{I}(\boldsymbol{w}_t; y_t)$, but we omit those here[4]. Additional training details are in the Appendix.

**Results.** We highlight several noteworthy observations revealed by our RIRL framework: (1) Principal inattention to individual outputs $\boldsymbol{z}$ and inattention to efforts $\boldsymbol{e}$ have distinct, nearly opposite, consequences on actor utilities; (2) Agents respond to the incentive structures created by Principal inattention to effort, with parallels to a prisoner's dilemma; (3) The temporal patterns of Principal attention reflect the value of information over time.

**Implications of Inattention Differ by Agent Ability** We analyze welfare related to actors' utilities. At equilibrium, the Principal and Agent utilities are negatively correlated (Figure 4a,b), comparing across

---

[4]We do add entropy regularization over $\omega(a|y)$ for both the Principal and Agent to encourage exploration.

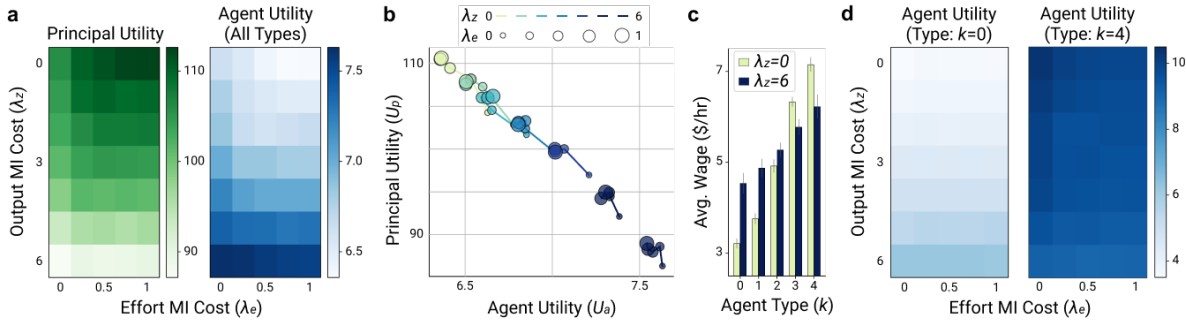

Figure 4: Sequential, multi-agent experiment results. All results are averaged over 20 runs and generated with $T = 5$. **(a, b)** Principal and Agent reward heatmaps (a) and scatter plot (b), for each $\lambda^z$ and $\lambda^e$. **(c)** Average wage for each Agent type, under a rational ($\lambda^z = 0$, yellow) and a boundedly rational ($\lambda^z = 6$, blue) Principal. Error bars denote to 95% confidence intervals. **(d)** reward for the lowest ($k = 0$) and highest ($k = 4$) ability Agent types, for each $\lambda^z$ and $\lambda^e$.

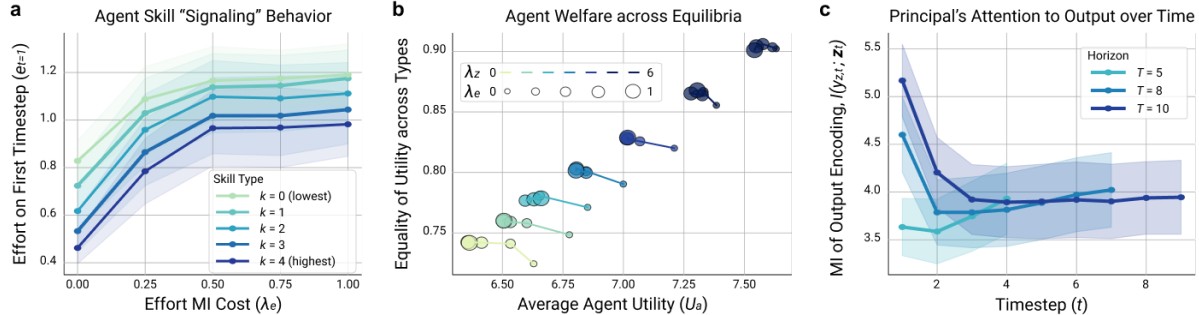

Figure 5: Additional sequential, multi-agent experiment results. All results are averaged over 20 runs. Shaded regions denote to 95% confidence. All results except (c) were generated with $T = 5$. **(a)** Average effort at $t = 1$ for each Agent type. **(b)** Equality across Agent types vs. Agent reward, for each $\lambda^z$ and $\lambda^e$. **(c)** Output information $\tilde{I}_{f^z}(y_t^z; \boldsymbol{z}_t)$ encoded over time (when $\lambda^z = 3$).

varying levels of $\lambda^z$ and $\lambda^e$. Note that the "rational" model has $\lambda^z = \lambda^e = 0$. This indicates that the Principal's bounded rationality has opposing implications for the Principal and Agents. From Figure 4a, we see that Agents' average reward increases and the Principal's reward decreases when the Principal's attention cost for individual outputs $\lambda^z$ increases. Conversely, Agent reward decreases with increasing Principal attention cost on effort $\lambda^e$. The cause for this is as follows. The Principal has a different optimal wage for each ability $\nu$ as shown in Figure 4c. When the Principal is fully rational it can use output and effort to accurately infer ability. Similar to what was seen in the bandit setting (above), increasing attention costs (in this case, on output $\lambda^z$) lead the Principal to set wages in a manner that is less profitable but also less attentionally demanding. At the resulting equilibria, *the Principal has more uncertainty over each Agent's type and adopts a "better safe than sorry approach" and increases the average wage to ensure output.*

However, while the *average* Agent reward increases with $\lambda^z$, it does not increase for all Agent types. Specifically, the reward of the (highest) lowest-ability Agent's (decreases) increases. Hence, the Principal's uncertainty over individual outputs decreases the wage (and reward) differences between Agents of different ability (Fig. 4c,d). This is particularly relevant when considering the *equality* of reward. A common inequality metric is the Gini coefficient (Gini, 1971), computed as a normalized sum of income differences; equality can be defined as `eq` $= 1 - \frac{N}{N-1}$`gini`. Figure 5b shows that Agents' average reward *and* equality across types both increase for higher output attention cost $\lambda^z$.

**Bounded Rationality induces Social Dilemma Dynamics.** This setup gives rise to interesting temporal strategic behaviors akin to *signaling* (Spence, 1973). Note that, while effort $e$ increases an Agent's

work disreward without increasing its income, Agents may still choose to exert non-zero effort $e > 0$. To provide an intuition for this, first, note that spending effort increases output rate ($z$ per hour $h$), resembling higher ability $\nu$. Second, the profit-maximizing wage $w_i$ increases with Agent ability[5] $\nu_i$. The Principal can't see each $\nu_i$ but can see the output $z_i$ and hours $h_i$. Hence, output rate is a "signal" of ability and Agents can misrepresent their ability by spending effort.

When $\lambda^e$ increases, it becomes costly for the Principal to distinguish between the Agent's ability and effort-action. Interestingly, this leads to *signaling equilibria* in which Agents choose to use more effort (Fig. 4a), resulting in lower average Agent reward and higher Principal reward (Fig. 5a). In effect, this Agent behavior has parallels to the "defect-defect" equilibrium in the prisoner's dilemma (Shoham & Leyton-Brown, 2008). When higher-ability Agents are not using effort, lower-ability Agents are incentivized to use effort to misrepresent themselves as having higher ability. This incentivizes the higher-ability Agents to also use effort to distinguish themselves from the lower-ability Agents. Figure 5a shows this effect: effort increases with $\lambda^e$ at all ability levels. In effect, increasing $\lambda^e$ creates equilibria where the Principal enjoys higher Agent effort essentially for free. Interestingly, the effects of effort being costly to observe are nullified if the individual outputs are also hard to observe (Fig. 5b), showing these two cost parameters interact.

**RIRL-actors Learn the Time-Value of Information.** We also show that RIRL can discover the time-value of information-acquisition and how it depends on the time horizon. Figure 5c shows output information that the Principal encodes $\tilde{I}_{f^z}(y_t^z; \mathbf{z}_t)$ over time. There is an initial spike in the amount of information at the beginning of the episode and this decreases over time. Intuitively, it is most efficient for the Principal to pay attention to outputs at the beginning of the episode, as this information can be used throughout the episode. Additionally, the initial spike is larger for longer horizons, as initial information has more value.

**Limitations.** While RI is a general model of boundedly rationality, it does not cover all models of human irrationality. For simulations that wish to use more specialized models of bounded rationality, RIRL may not be sufficient. Examples include hyperbolic discounting (Kirby & Herrnstein, 1995) to model time-inconsistent delay discounting (humans tend to prefer rewards in the near future much more than rewards farther in the future) or prospect theory (Tversky & Kahneman, 1992) to model loss-aversion. Future research may extend RIRL to include such notions of bounded rationality.

# 6 Conclusion

We propose a novel framework, RIRL, for modeling bounded-rationality in MARL simulations with complexity beyond the scope of prior techniques. Our method incorporates *Rational Inattention*, an established model of human bounded rationality which uses mutual information to model the cognitive cost of processing information. Mutual information has been used extensively in RL for exploration and skill discovery, but, to our knowledge, we are the first to incorporate it for modeling human-like behavior in multi-agent simulations. We evaluate our method in two Principal-Agent problem settings, including a complex multi-agent setting, with multiple information channels with heterogeneous costs. Incorporating bounded rationality leads to different actor strategies and welfare outcomes as compared to under rational assumptions. These results establish RIRL as a promising framework for using MARL to analyze systems of human agents.

# 7 Ethics and Societal Impact Statements

Our work proposes a framework to model bounded rationality in Multi-Agent Reinforcement Learning based simulations. Thus our framework may be used to draw implications in simulations modeled after real-world systems. While addressing the rationality gap between human actors and RL actors is an important step towards this goal, there is still much more development needed for achieving the realism required for real-world decision making based on AI simulations. We do not intend our framework to be used to explore methods to increase discrimination or unfairness in real-world systems, instead for it to be used to investigate how to decrease such biases. For our work, and in general for agent based simulations works, clearly stating the current limitations of the models used helps mitigate this issue.

---

[5]This is a simple consequence of how reward functions are defined. See Fig 4c for confirmation.

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

## A  Additional Bandit Experiment Details and Results

### A.1  Additional Details

**Noise Structure.**  Recall in the bandit experiments, the output $z$ is a noisy function of the hours action $h$ chosen by agent. Specifically we use the following output distribution:

$$p(z|h) = \begin{cases} 0.7 & \text{if } z = h \\ \frac{0.7^{|z-h|}}{0.3 \cdot c} & \text{otherwise} \end{cases} \tag{19}$$

Where $c$ is a numerically calculated factor such that $p(z|h)$ is a proper probability distribution satisfying $\sum_{z \in \mathcal{Z}} p(z|h) = 1$. This noise distribution assures high correlation between $h$ and $z$ with the probability of $z$ levels decreasing exponentially as the difference between $h$ and $z$ grows. We plot the probability heatmap in Figure 6a.

**Constant Relative Risk Aversion (CRRA) Function** The constant relative risk aversion (`crra` function (Pratt, 1978)) commonly used in economics has constant risk aversion which means decisions are invariant to scale. It is a concave function where the risk aversion parameter, $\rho \geq 0$, determines the concavity. This models diminishing returns with higher values. The function is given by:

$$\text{CRRA}(u, \rho) = \begin{cases} \frac{u^{(1-\rho)}-1}{1-\rho} & \text{if } \rho \neq 1, \rho \geq 0 \\ ln(u) & \rho = 1 \end{cases} \tag{20}$$

**Inattention Model From Theory.** The model studied in classical theoretical Principal-Agent literature we compare with (Mirrlees, 1976) utilizes the same reward functions for the principal and the agent. It has a slightly different model of principal inattention where it models costly principal inattention with a factor $\theta$ that controls the magnitude of the observation noise in the output signal. Specifically, for a true output level $z^*$, the principal observes output $z = z^* + \frac{1}{\theta}\epsilon$ where $\epsilon$ is noise. The principal incurs an attention cost that is a function of $\theta^2$ and is subtracted from their reward function. While this attention model is different from our mutual information based model, it shares many similar properties. In both our models the inattention introduces noise which makes acting optimally difficult. Additionally in both models, an attention parameter can reduce this noise at a cost to the principal and the principal can choose the optimal, reward-maximizing value of this attention parameter. Such similarities explain why, even under different models, our results are similar to those from theoretical analysis.

**Training Hyperparameters.** We used learning rates of 1e-3 for the training the principal policy parameters and 5e-3 for the mutual information classifier. We used a batch size of 128 and trained the principal for a total of 100000 batches. During training we gradually annealed $\lambda_{a_p}$ from 0 to the desired value at a rate of 4/10000 per batch. We average all results across 5 random seeds and we set the random seed for pytorch, numpy, and python's internal random module for each run. We used seeds of $[0, 4]$. All experiments were run on 16CPU cloud compute machines with 54GB of memory. Given a pay schedule which consists of mean and standard deviation parameters $(\boldsymbol{\mu_z}, \boldsymbol{\sigma_z})$ for each output level $z \in \mathcal{Z}$, to calculate the principal policy we first sample 100 pay schedules and calculate the agent reward per output for each output level for each pay schedule. We then average to calculate the average agent reward for each output level. We use the noise structure to calculate the average reward for each action and use the soft-q formulation over the utilities given in the main text to obtain the agent's stochastic policy.

### A.2 Additional Results

We provide a few additional figures for the bandit experiment. Recall the principal's policy was the pay schedule parameterized by mean and standard deviation values for each output level $(\boldsymbol{\mu_z}, \boldsymbol{\sigma_z})$. Figure 6b shows the standard deviation parameters $\boldsymbol{\sigma_z}$ learned under MI and Entropy regularization for different cost factors $\lambda$. Increasing entropy regularization results in very large increases to the standard deviation parameters while MI regularization leads to much smaller standard deviation increases. The pay schedule means for Entropy regularization become sharper to overcome this increase in standard deviation.

Figure 6c compares pay schedule means $(\boldsymbol{\mu_z})$ across different Agent policy temperature $(\beta)$ values. For low inattention costs, the pay schedules are similar. However the pay schedules for higher $\beta$ (more deterministic agents) decrease slower with increasing principal attention cost. Given the pay schedule parameters, the agent has an optimal $h$ action. More deterministic agents take this optimal action with higher probability so it is more valuable for the principal to incentivize them to higher optimal actions, which in turn lead to higher outputs. This explains why Agent and Principal reward are higher for higher $\beta$ values.

## B   Additional Multi-Agent Multi-Timestep Experiment Details and Results

### B.1   Additional Details

**Training Hyperparameters.** We used learning rates of 1e-4 for the Principal and Agent's policy parameters and 1e-3 for all the mutual information classifiers. We used a batch size of 512 episodes and train the

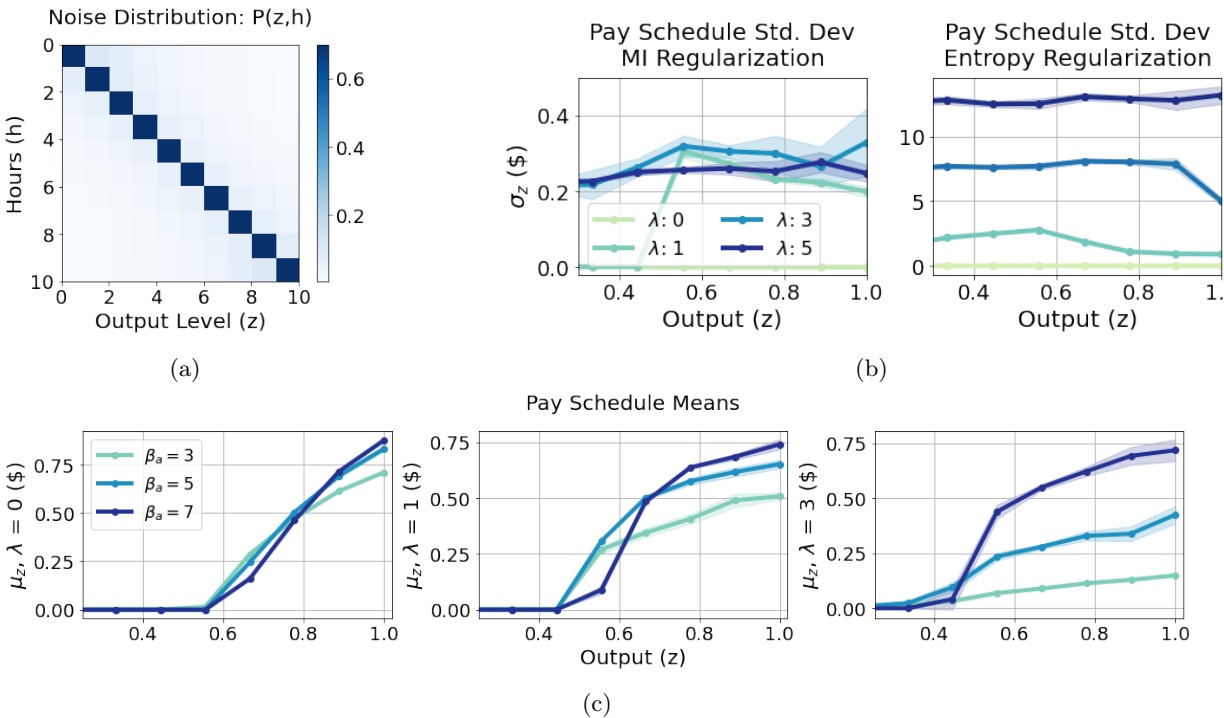

Figure 6: Additional bandit experiment results. All results were averaged across 5 random seeds and have 95% confidence regions shaded. (a) The noise distribution between hours and output ($P(z|h)$). (b) The pay schedule standard deviation plots for MI and Entropy Regularization. (c) A comparison of pay schedule means across different Agent policy $\beta$ values.

principal and agent through 60000 batches. We train a single RIRL-actor for the Agents and concatenate the experiences of all $n_a$ Agents when updating the policy. The Agent policy therefore effectively has a batch size of $512n_a$. To avoid vastly different total episode returns, we scaled the rewards by the horizon during training. We run all experiments on 8CPU cloud computing machines with 26GB of memory. We average all results across 20 random seeds and we set the random seed for pytorch, numpy, and python's internal random module for each run. We used seeds of $[0, 19]$.

## B.2 Additional Results

We present some additional results for the sequential multi-agent experiments. Figures 7a,7b show Principal reward with output MI cost ($\lambda^z$) and effort MI cost ($\lambda^e$) for different horizons. We see the trend of Principal reward decreasing with $\lambda^z$ and increasing with $\lambda^e$ occurs across all horizons.

Figure 7c shows unscaled MI cost for output and effort with $\lambda^z$ and $\lambda^e$. We see that $I(y_e|e)$ decreases with both $\lambda^z$ and $\lambda^e$. $I(y_z|z)$ decreases with $\lambda^z$ but is not affected greatly by $\lambda^e$.

Figure 7d plots the agent reward for all agent types. It expands the plot of Figure 4d. The plots show the difference between the reward for each agent type and the average reward.

Figure 7e demonstrates how different values for different agent types change with output MI cost ($\lambda^z$). We plot each result as the change from the $\lambda^z = 0$ values. As mentioned previously, reward increases with $\lambda^z$ for lower ability agents while it decreases for the highest ability agent. We observe similar trends for hours $h$. We additionally notice effort $e$ generally decreases which leads to $z$ staying constant or decreasing for all agent types, even though some agent types have increased hours. This lower output along with higher average wages explains the decrease in Principal reward.

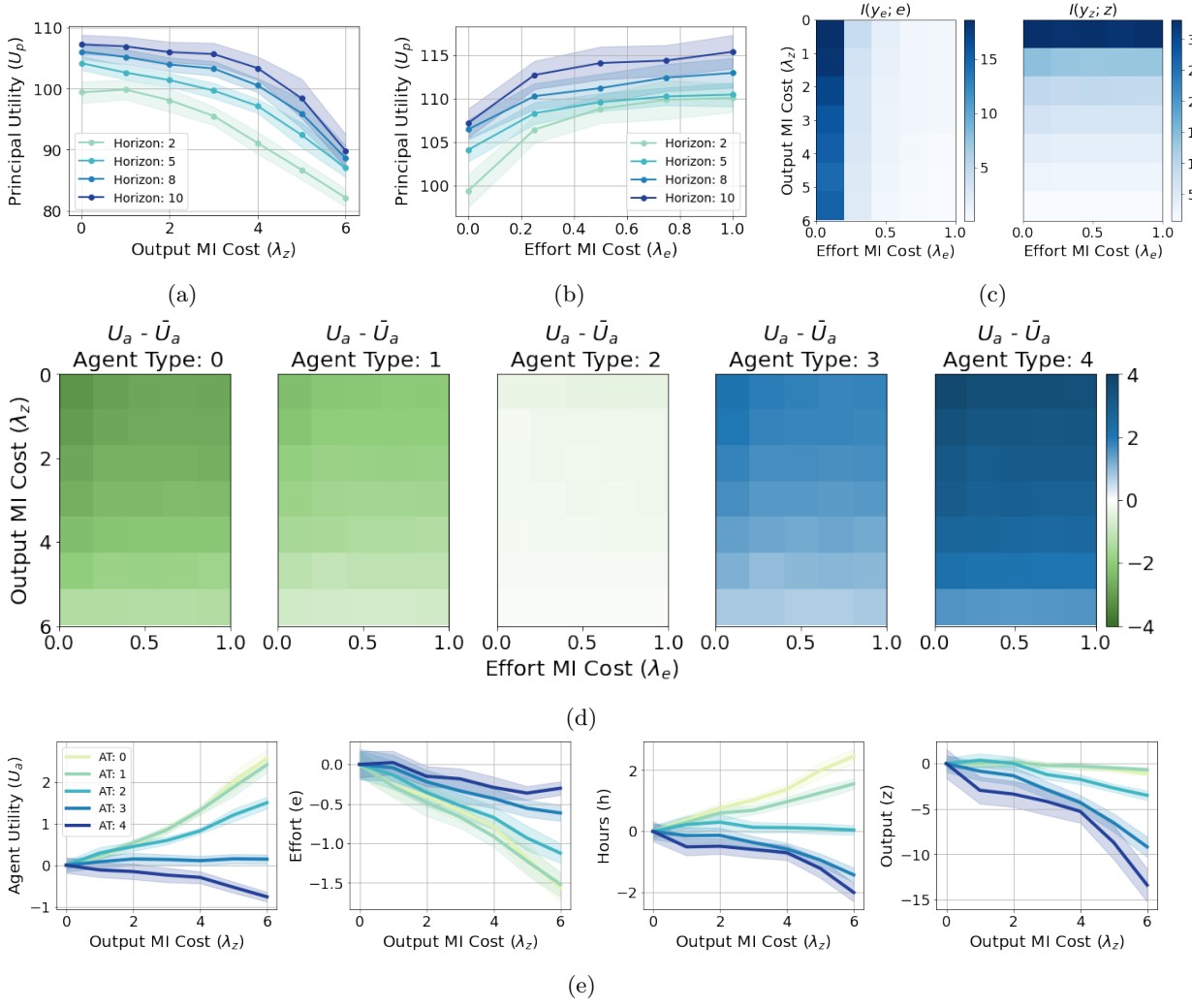

Figure 7: Additional sequential, Multi-Agent experiment results. All results are averaged over 20 random seeds and all confidence regions depict 95% confidence intervals. (a, b) Principal reward with Output ($\lambda^z$) and Effort ($\lambda^e$) MI Costs. (c) The unscaled Output and Effort MI with $\lambda^z$ and $\lambda^e$. (d) The different between each agent type's reward and the mean reward across all types with $\lambda^z$ and $\lambda^e$. (e) How agent reward, effort, hours, and output changes with respect to Output MI Cost $\lambda^z$ for each agent type. All values are plotted as the change amount from the $\lambda^z = 0$ value.

## C    RIRL Implementation Details

We use this section of the appendix to cover important implementation details and tips. This is intended for practical guidance.

**Stochastic Encoder Module Configuration and Initialization.**    Section 4 describes the architecture of our RIRL-actor policy class. As described, we learn an encoder $f^m(y_t^m|o_t^m, h_t)$ for each observation channel $m$. Encoder $f^m$ takes observation $o_t^m$ and recurrent state $h_t$ as inputs and outputs the parameters (means and standard deviations) of a stochastic encoding $y_t^m$.

We find that, in practice, learning does not progress if each $y^m$ contains very little information about $o^m$ at the start of training. To address this, we recommend two implementation choices. First, implement $f^m$ as

a residual-style module:

$$\mu_t^m, \sigma_t^m = f^m(o_t^m, h_t), \quad y_t^m = o_t^m + \mu_t^m + \sigma_t^m \cdot \epsilon_t^m, \tag{21}$$

This simply requires setting the output $y^m$ to have the same size as observation $o^m$ and adding $o_t^m$ to the mean $\mu_t^m$. Second, initialize the output layer of $f^m$ such that $\sigma^m$ is consistently very small. We perform this by adding a constant negative offset to the bias units associated with the $\log \sigma^m$ outputs, which we exponentiate to get $\sigma^m$. As a result of this strategy, $y_t^m$ closely follows $o_t^m$ at the start of training.

**Hidden State as an Encoder Input.** We emphasize that the inputs to encoder $f^m$ is the concatenation of the observation $o_t^m$ and the hidden state $h_t$. Similarly, when using the discriminator $d^m(y_t^m, [o_t^m, h_t])$ to estimate $\tilde{I}_{f^m}$ and when training the discriminator, we also apply this concatenation. In other words, $d^m$ regards $[o_t^m, h_t]$ as the observation, such that the $\tilde{I}_{f^m}$ captures the MI between $y_t^m$ and $[o_t^m, h_t]$.

This is an important detail for ensuring that MI regularization works as expected in the multi-step setting. For instance, we observed that, if the discriminator does not see the hidden state $h$, $f^m$ learns to encode $o^m$ such that $I(y^m; o^m)$ is minimal but where $o_t^m$ can still be easily recovered given $y_t^m$ and $h_t$.

**Optimization** During training, we found that learning was most stable if separate learning rates were used for the policy modules $\{f^1, \ldots, f^M, \text{LSTM}, \omega\}$ and for the discriminator modules $\{d^1, \ldots, d^M, d^\omega\}$. Importantly, *the discriminator modules use a 10x higher learning rate.* Concretely, we use learning rates of 0.0001 and 0.001 for the policy and discriminator modules, respectively. Configuring learning rates this way helps to ensure that discriminator $d^m$ can adjust to changes in encoder $f^m$ faster than the encoder can adapt to changes in the discriminator. Intuitively, this improves the quality of the MI estimates during training.

Another important optimization detail concerns gradient flow. Gradients from $\nabla \log \omega(a_t | \cdot)$ need to back-propagate through the encodings $[y_t^1, \ldots y_t^M]$ in order for the encoder modules to receive meaningful gradients. In Pytorch, which we use for this implementation, ensuring that this gradient flow occurs requires some attention. Given the (learned) mean and standard deviation parameters (which are functions of the encoder input), Pytorch constructs the sampling distribution as `m = Normal(`$\mu, \sigma$`)`. The *output* of the encoder module $y_t^m$ must be sampled via the reparameterization trick: `y = m.rsample()`, which allows gradients to flow through $y_t^m$. Finally, this output should be detached from the backpropagation graph when calculating its *log probability*: `encoder_log_prob = m.log_prob( y.detach() )`, which is needed for computing the policy gradients. Similarly, care must be taken to ensure that inputs to the discriminators are detached in the same way.

