# OpenReview forum: "Modeling Bounded Rationality in Multi-Agent Simulations Using Rationally Inattentive Reinforcement Learning"
_TMLR — Accepted by TMLR_

### Review · Reviewer_H3aW · 2022-10-07

**Summary Of Contributions:**

This article proposes a new multiagent reinforcement learning algorithm to model bounded rationality by means of a Lagrangian term to the reward function. The particular form of the term is a mutual information between the relevant features (observation) and the actions taken by the agent. This term has an interpretation of modelling rational inattentiveness from economic theory. Authors briefly contrast this to a traditional entropy cost regularisation. The article presents a comparison with analytical models in a simple Principal-Agent setting where exact solution are known, and extend the setting to sequential decision making and multiple observational channels, each with its own attention cost. The authors draw some interesting conclusions and show how the extensions provide a more realistic sets of behaviours for both principals and agents.

**Broader Impact Concerns:**

Whenever a technology can be used to model humans, there might be a risk that it can be abused. In this case, the risk is minimal as the tools can be better used for pure understanding than manipulation. No salient ethical concerns exist.

**Requested Changes:**

Although the authors cite a good number of bounded-rationality articles, there are some important omissions that would enhance the framing of the article:

* Nick-Marios T. Kokolakis, Aris Kanellopoulos & Kyriakos G. Vamvoudakis. "Bounded Rationality in Differential Games: A Reinforcement Learning-Based Approach": This article models bounded rationality as a level-k policy instead of full Nash equilibria.
* Erevl, Ido, and Alvin E. Roth. "Simple Reinforcement Learning Models and Reciprocation." Bounded rationality: The adaptive toolbox (2002): 215. This article uses RL to model bounded rationality agents in iterated prisoners dilemmas, extending prior analytical models in game theory.
* Hernandez, Jose G. Vargas, and Ricardo Perez Ortega. "Bounded rationality in decision–making." MOJ Research Review 2.1 (2019): 1-8.
* Duéñez-Guzmán, Edgar A., et al. "Statistical discrimination in learning agents." arXiv preprint arXiv:2110.11404 (2021). This article uses RL agents to model bounded rationality, and explicitly addresses information processing of agents to explain their behaviour.

Please add more detail to the estimation of mutual information, and how biased is it.

**Strengths And Weaknesses:**

# Strengths

The article is reasonably well written and provides an important extension to the modelling of bounded rationality using reinforcement learning agents. The use of an explicit term to model the difficulty of incorporating observed information into a policy is novel to my knowledge, and provides an interesting way to study bounded rationality explicitly rather than implicitly.

The generalisation to multiple possible observation channels, each with their own intrinsic attention cost makes this even more flexible to model human cognitive processes.

The results of the Principal-Agent models, particularly when compared against analytical models, and then extended to more complex and analytically intractable variants is well presented and explained. I found it interesting to see the responses of costs of rationality on the decisions made, and hence on the utilities received.

# Weaknesses

Reinforcement learning has been used to model bounded rationality in a wide variety of situations. The assumption that RL agents are perfect optimisers doesn't hold in many realistic circumstances. It is a disservice to the article to frame it like this, because the present work provides a good way to _explicitly_ model the bounds of rationality, rather than having those bounds be _implicit_ like it has been done in prior literature.

The connection to an entropy cost regulariser should be brought up earlier. It is strange to have it be the basic baseline for comparison when it has not been discussed before section 3.1.

The mixed use of reward and utility can be confusing. It would be better to stick with reward / return, and just mention that in the economics literature it tends to be referred to as the utility (or the other way around).

The discussion on mutual information estimation should provide more details. For instance, the authors define an unbiased estimator, but then don't address whether their montecarlo estimation is unbiased or nor. In fact, I would suspect that the estimation is highly biased as it comes from rollouts of the current policy, where the $\log p(o)p(a)$ samples are simply computed from across the batch, and $\log p(a, o)$ comes from along a single rollout. How does this estimation compare to the real mutual information? The authors claim that even a single sample is good in practice, but no analysis is provided.

The authors suggested that the mutual information should take into account the hidden information in section 4.2.1. However, later on, in their actual examples, the authors explicitly state that in sequential decision making it is particularly advantageous for the Principal to expend attention early in the episode and have that information available for the rest of the episode. This seems to be in conflict,  am I missing something?

---

> ### Author Response · Authors · 2022-10-25
> **Response**
>
> Thank you for your review!
>
> **Related work**
>
> Thank you for suggesting the related work, we will include this.
>
> **Reinforcement learning has been used to model bounded rationality in a wide variety of situations. The assumption that RL agents are perfect optimisers doesn't hold in many realistic circumstances. It is a disservice to the article to frame it like this, because the present work provides a good way to explicitly model the bounds of rationality, rather than having those bounds be implicit like it has been done in prior literature.**
>
> Thank you for pointing this out, we will update the language to reflect this idea that RL agent don’t always find/learn the (theoretically) optimal policy.
>
> **The connection to an entropy cost regulariser should be brought up earlier. It is strange to have it be the basic baseline for comparison when it has not been discussed before section 3.1.**
>
> Thank you, yes, we can introduce this comparison earlier in the paper.
>
> **The mixed use of reward and utility can be confusing. It would be better to stick with reward / return, and just mention that in the economics literature it tends to be referred to as the utility (or the other way around).**
>
> Thank you for pointing this out, we will use the terminology “reward” throughout.
>
> **The discussion on mutual information estimation should provide more details. For instance, the authors define an unbiased estimator, but then don't address whether their montecarlo estimation is unbiased or nor. In fact, I would suspect that the estimation is highly biased as it comes from rollouts of the current policy, where the log⁡p(o)p(a) samples are simply computed from across the batch, and log⁡p(a,o) comes from along a single rollout. How does this estimation compare to the real mutual information? The authors claim that even a single sample is good in practice, but no analysis is provided.**
>
> First, note that we train a discriminator to estimate the *ratio* p(a, o) / p(a)p(o), not the individual probabilities itself, which is easier.
>
> Second, the policy $\pi$ typically changes slowly during training. Hence, there is only a small amount of distribution shift in the $(a, o)$ samples across training steps. So using a single example could be high *variance*, but training still works when training the discriminator slowly enough.
>
> Third, the pair $(a, o)$ of a randomly sampled $a$ and a randomly sampled $o$ can be taken from a batch of trajectories that were collected under different $\pi$s, because such randomly sampled pairs are more likely to be independent across trajectories as the policy changes.
>
> As such, if we sampled many trajectories from the same policy $\pi$, and given a discriminator with enough model capacity, it should have the capacity to learn the true odds-ratios. With a smaller number of samples, training the discriminator needs to be done slowly enough, but in practice, our approach is sufficient to learn meaningful MI estimates, i.e., that sufficiently distinguish independent vs dependent samples $(a, o)$.
>
> **The authors suggested that the mutual information should take into account the hidden information in section 4.2.1. However, later on, in their actual examples, the authors explicitly state that in sequential decision making it is particularly advantageous for the Principal to expend attention early in the episode and have that information available for the rest of the episode. This seems to be in conflict, am I missing something?**
>
> In 4.2.1, the hidden state refers to the recurrent neural network’s state that’s carried over over time.
> In spending attention, “information” refers to observing the world state more, and not the recurrent neural network’s hidden state.
> The terminology “hidden” is standard in deep learning, but is confusing here. We will remove “hidden” from 4.2.1 to avoid this confusion.

---

> > ### Comment · Reviewer_H3aW · 2022-11-14
> > **Thank you for the response**
> >
> > Thank you for the answers. And thanks for clarifying the computations of the MI. You are right, that this would be high variance, not bias. I still would like to see some supporting work on the comparison between single-sample and more samples, as all these points are made purely based on intuition. By providing the data / analysis, you would train the intuition of the readers, and allow them to develop their own version in a better informed manner.

---

### Review · Reviewer_pDCi · 2022-10-12

**Summary Of Contributions:**

This paper considers the incorporation of human irrationality into reinforcement learning models (Rational Inattention model), which has the potential to improve traditional RL models that assume agents behave rationally. The authors propose a rationally-inattentive objective and extend it to allow for multi-step dynamics and information channels with heterogeneous costs. The authors also follow previous literature for practical estimation of mutual information. Experiments are conducted based on a bandit setup and a multi-agent scenario.

**Broader Impact Concerns:**

I do not have significant concerns on the broader impact statement section.

**Requested Changes:**

- Experiments: I think the paper can be improved by evaluating the method on more complex tasks (e.g., StarCraft II micromanagement tasks) besides the domains considered in the paper.

- Writing: I think it would improve the paper by elaborating on more details for Equation 6 and Equation 7 (extension of the method to support multiple channels, etc). It is not very straightforward to understand these equations.

- Discussion about entropy-regularized RL objectives: It would be interesting to also discuss the connection of the rationally-inattentive objectives and entropy-regularized objectives (which can be considered as motivating the policy to be uniform, quite similar to what is done here-ignoring part of the observations).


**Strengths And Weaknesses:**

This paper incorporated human irrationality into reinforcement learning models. The authors motivate the need for such incorporation well, and propose a corresponding rationally-inattentive objective with extensions to more practical setups. The authors also propose a practical implementation of the objective following the literature.

- Strengths
    - Originality: The setup seems quite original - incorporating rationality concerns to multi-agent RL, although the way to estimate mutual information follows previous literature.

    - Quality and clarity: The paper is well-written - the authors motivate the need to incorporate rationality to RL agents, and propose a rationally-inattentive objective with a practical implementation of the method.

- Weakness
    - Significance: My main concern for the paper is in its experimental evaluation part. I acknowledge the experiments the authors conducted. However, the bandit setting is not practical enough. In addition, there are a number of well-established MARL benchmarks (multi-agent particle, StarCraft II micromanagement tasks). Can the method scale to the more practical and complex tasks (which may need to revise the environment)?

---

### Review · Reviewer_6GV3 · 2022-10-13

**Summary Of Contributions:**

This paper is about modeling bounded rationality in multi-agent reinforcement learning (MARL). RL is often modeled as an MDP problem. So, the objective of RL, to maximize the cumulated rewards, requires agents to be rational, which is not human-like. To this end, this paper tries to build human-like agents with bounded rationality.

Specifically, the authors propose Rational Inattention Reinforcement Learning (RIRL), where bounded rationality in this paper is modeled by mutual information. The intuition is that the agent should take actions that are irrelevant to current observations. Such bounded rationality is used as an additional reward/cost in the RL formulation. The authors propose several channels, i.e., I(o;a), I(y;o)+I(a;y), I(y;o)+I(a;[y1,...yM]), where y is obtained by VAE. Besides, due to the challenges to compute mutual information, the authors use a discriminator to estimate mutual information.

The experiments are based on Principal-Agent problems with a bounded rational Principal.  RIRL is evaluated in two scenarios. One is involved with only one Principal and one Agent. Another is involved with one Principal and a team of Agents. The results show the emergent behavior of RIRL.


**Broader Impact Concerns:**

No further concerns about the ethical impact.

**Requested Changes:**

- Discussion about the related work, especially the difference with [2].
- More experiments in more scenarios except for Principal-Agent problems.

**Strengths And Weaknesses:**

Pros
- Incorporating human-like rationality into RL is an interesting problem.
- The paper is generally well-written and easy to follow.
- The emergent behavior experiments are interesting, especially the multi-agent experiments.

Cons
- The idea of modeling bounded rationality with mutual information is not novel. In [2], as Eq.8 shows, mutual information has been used.
- Related work is not sufficient. Maybe the authors should discuss [1,2]
- The single-agent experiment is a toy example. Because the paper mainly discusses MARL, while the single agent experiment degrades to a bandit situation.
- The empirical results, though interesting, are not complex enough.
- Human-like behavior modeling is important to the MARL community, but I am not sure whether Principal-Agent problems can be the main part of human-like behavior modeling. Maybe the authors should consider more scenarios.

Minor
- In Eq.13, e_i is sampled from \pi(e|W), however W is a function. The authors should clarify how this soft-Q policy takes a function as one parameter.
- Why stochasticity can be modeled as a Gaussian distribution for the wage distribution?

[1] Herbert A Simon. Theories of bounded rationality. Decision and organization, 1(1):161–176, 1972

[2] Tim Genewein, Felix Leibfried, Jordi Grau-Moya, and Daniel Alexander Braun. Bounded rationality, abstraction, and hierarchical decision-making: An information-theoretic optimality principle. Frontiers in Robotics and AI, 2:27, 2015.

---

### Public Comment · ~Tailia_Malloy1 · 2022-09-24
**Potentially relevant reference**

In the section on "Mutual Information in Reinforcement Learning" of the related works section a claim that "with no prior work, to our knowledge, considering the domain of multi-agent simulations." Our paper "Capacity-Limited Decentralized Actor-Critic for Multi-Agent Games" IEEE CoG 2021 may be of interest as a potential reference if the authors agree. In that paper we apply a penalty to reward based on policy mutual information in a multi-agent decentralized actor, centralized critic model based on an alteration of the MADDPG method. The rationally-inattentive objective described in this work is similar in motivation to what we describe as the 'capacity-limited' learning objective, which functions by penalizing reward based on a weighted estimate of policy mutual information. While not functioning exactly the same they are similarly motivated and designed, and the multi-agent domain of our work may make it of interest as a reference. In that paper we report improved performance on the multi-agent particle environment set with the capacity-limited learning objective.

---

> ### Author Response · Authors · 2022-10-01
> **Thank you for the reference**
>
> Thank you for pointing us to your reference! It was great to read and definitely relevant. We will include it in the related works in a later version of our paper. Just for the sake of clarity, we will note some differences in this comment: while we both consider limited information in multi-agent settings, the settings we consider (modeling bounded rationality for more realistic MARL simulations, versus, a method for policy regularization to learn policies with better performance) are very different. As a result, as mentioned in our paper, our methods and policy class are very different from this work and other prior works in reinforcement learning that use mutual information as regularization (ex Leibfried et al. 2020). For example, our policy class considers allowing a rich set of behavior to be modeled, such as different attention costs on different parts of the observation space.  Additionally, due to the difference in settings, our method also draws different conclusions, focusing on the implications of modeling human irrationality such as which agents benefit and how metrics, like equity, evolve.

---

### Decision · Action_Editors · 2022-11-25

**Recommendation:** Accept with minor revision

**Comment:**

This paper presents a rationally-inattentive RL learning that uses mutual information as an objective function. The paper compares the approach to the entropy regularization typically used by RL practitioners during training. There are applications of the idea to several different settings (within Principal-Agent problems); the results are interesting and relevant to the community.

It remains to be seen how easily scalable these ideas are, and several reviewers pointed the lack of evidence on larger domains as one drawback of the paper. However, after some discussion, reviewers agreed that this does not significantly affect the claims substantiated by the paper, and that the paper ultimately satisfies the review criteria described in TMLR's guidelines.

The main problems with the paper have been identified by reviewers, partly due to the vast prior work on these topics:
- There is only modest novelty. Any clarifications that can be made to highlight the novelty from past work is encouraged.
- There are also missing citations and discussion of related work.

Please take the review feedback into consideration when finalizing the paper.

**Audience:**

Yes, very much. The paper is about modeling bounded rationality in multiagent RL using mutual information that is estimated via discriminators (inspired by GANs).

**Claims And Evidence:**

Yes.

The reviews stated the empirical demonstration is in a rather simplified single-agent bandit setting, calling the claims into question, but that is actually not the case. Section 5.2 shows a multi-agent sequential setting, and emergence of social dilemma dynamics.

The claims are well-supported.

---

> ### Author Response · Authors · 2022-12-14
> **Thank you**
>
> Thank you for chairing this submission and your decision. We've uploaded a camera-ready version.